# Putting a spin on metamaterials: Mechanical incompatibility as magnetic frustration

Ben Pisanty,[1, *] Erdal C. Oğuz,[2, 3, 4, †] Cristiano Nisoli,[5, ‡] and Yair Shokef[2, 4, 6, §]

[1]School of Phyics and Astronomy, Tel Aviv University, Tel Aviv 69978, Israel
[2]School of Mechanical Engineering, Tel Aviv University, Tel Aviv 69978, Israel
[3]School of Chemistry, Tel Aviv University, Tel Aviv 69978, Israel
[4]Sackler Center for Computational Molecular and Materials Science, Tel Aviv University, Tel Aviv 69978, Israel
[5]Theoretical Division, Los Alamos National Laboratory, Los Alamos, NM, 87545, USA
[6]Center for Physics and Chemistry of Living Systems, Tel Aviv University, Tel Aviv 69978, Israel

Mechanical metamaterials present a promising platform for seemingly impossible mechanics. They often require incompatibility of their elementary building blocks, yet a comprehensive understanding of its role remains elusive. Relying on an analogy to ferromagnetic and antiferromagnetic binary spin interactions, we present a general approach to identify and analyze topological mechanical defects for arbitrary building blocks. We underline differences between two- and three-dimensional metamaterials, and show how topological defects can steer stresses and strains in a controlled and non-trivial manner and can inspire the design of materials with hitherto unknown complex mechanical response.

## I. INTRODUCTION

Mechanical metamaterials are structured from mesoscopic building blocks, whose individual characteristics and mutual arrangements dictate global properties and functionalities, potentially leading to exotic macroscopic responses [1–4]. For instance, a pruning process selectively applied to random spring networks can cause them to approach either the incompressible or completely auxetic limits [5], as well as tune specific long-range coupled mechanical responses [6]. Hierarchical cut patterns in elastic media allow for extremely large strains and emergence of macroscopic shapes when stretched [7]. In lattice-based structures, defects and dislocations can localize collective soft modes [8] and guide folding motions [9].

Combinatorial metamaterials, realized by an array of soft or hinging anisotropic building blocks have elicited much recent interest [10–12]. The ability to control the orientations of individual blocks allows access to highly complex non-periodic designs, and may lead to soft compatible structures with advanced mechanical functionalities, such as mimicking kinematic mechanisms [1], textured sensing [10], or shape changing [13], with possible applications in pluripotent origami [14]. In such systems, only very specific arrangements of the building blocks lead to cooperative soft deformations. Most arrangements, however, contain multiple contradictions: contacting blocks tend to deform in opposing directions.

Mechanical incompatibility controls the stiffness of the metamaterial [10]. It also results in localized response to an external force and thus limits its functionality [11, 15]. Crucially, it can be harnessed for advanced functionalities such as multistability [16] and programmability [17].

In particular, deliberate incompatibility of the constituting units can lead to topological defects and to complex mechanical responses [11, 18]. Hence, understanding and manipulating mechanical incompatibility opens a path toward mechanical control at the macroscopic level. When considering the directions of deformations as binary arrows, the study of building block incompatibility in mechanical metamaterials can be greatly facilitated by an analogy with geometrically-frustrated lattices [19], random spin glasses [20] and spin-ice systems [21–26].

In this article, we introduce a general framework for identifying and generating topological defects due to mechanical incompatibility in metamaterials based on the analogy with frustrated spin systems, and provide guidelines for a material-by-design approach. Our formalism describes incompatibility via Wilson loop products [27], which count the parity of antiferromagnetic effective interactions among emergent pseudo-spins, in complete analogy to the case of geometric frustration in classical Ising spin systems [28]. Our spins, in turn, are related to mechanical deformations in the metamaterial. We apply this framework to a novel class of two-dimensional (2D) combinatorial mechanical metamaterials constructed of hexagonal building blocks as well as to three-dimensional (3D) metamaterials, whose compatible architectures have been investigated recently [10]. We demonstrate the capability of our approach to induce complex frustration motifs such as defect lines in 3D systems that can lead to twisted stress distribution in the material, or defect loops in 3D that can cause stress to concentrate in a certain region or alternatively to avoid that region, merely by controlling the texture of the boundary forcing.

## II. MAGNETIC SPIN ANALOGY IN MECHANICAL METAMATERIALS

As a particular example of our general strategy, consider the anisotropic hexagonal building block with hinging facets presented in Fig. 1(a). Its soft deformation mode, in which the constituent links do not change in

* benpisanty@mail.tau.ac.il
† erdaloguz@mail.tau.ac.il
‡ cristiano@lanl.gov
§ shokef@tau.ac.il; https://shokef.tau.ac.il

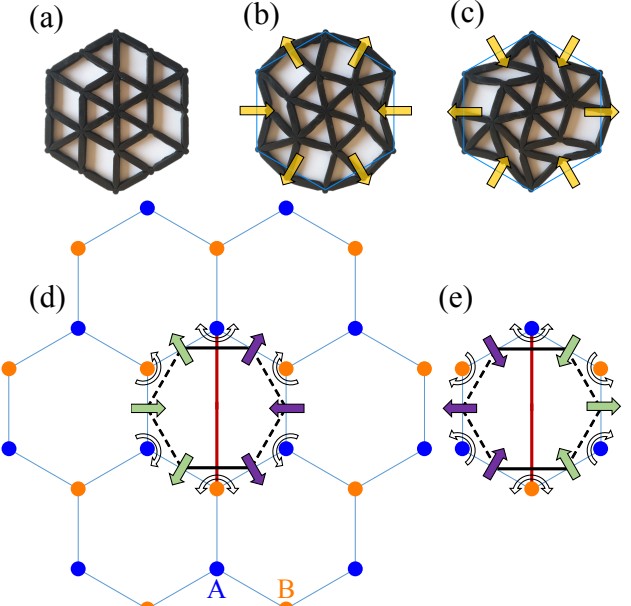

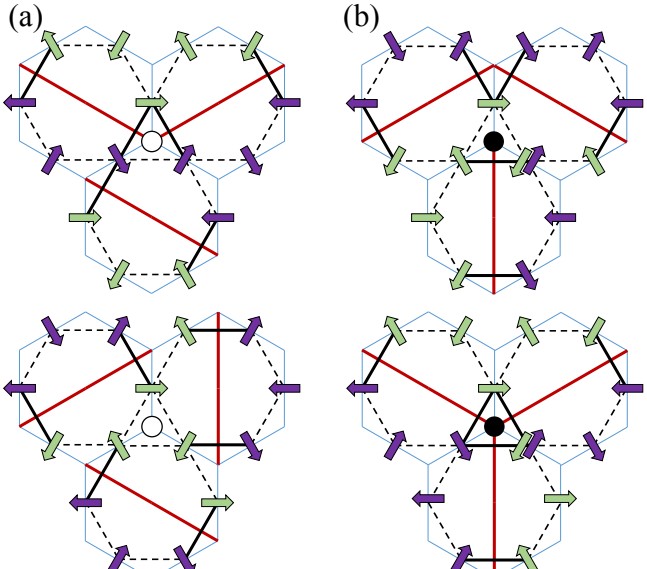

FIG. 1. (a) 2D hexagonal building block, (b) the extended 2-in-4-out, and (c) the contracted 4-in-2-out states of its soft deformation mode, which does not stretch or compress the constituent links. (d,e) Vertices of sublattices A and B are marked in blue and orange respectively. Ising spins are assigned to the deformation arrows according to the winding direction around the two sublattice vertices: ±1 spins are indicated in purple and green. Spin of two adjacent facets is preserved (flipped) if deformations wind in the same (opposite) direction with respect to the common vertex between them, as indicated by the circular arrows. Resulting ferromagnetic (antiferromagnetic) interactions are indicated by dashed (solid) lines connecting the two facets. Red director line drawn perpendicular to the antiferromagnetic bonds designates the orientation of the building block.

FIG. 2. (a) Compatible and (b) incompatible vertices respectively consist of an even or odd number of antiferromagnetic bonds on the corresponding triangular plaquettes of the dual kagome lattice. Black circle indicates a topological mechanical defect. The four depicted interfaces account for all the possible configurations of three hexagons meeting at a vertex, up to rotations and reflections.

length, consists of deformations along the six symmetry directions such that a 2-in-4-out or 4-in-2-out rule applies, as indicated by the yellow arrows, see Fig. 1(b,c). The rule reduces its symmetry from six-fold to two-fold, around a director line marked in red in Fig. 1(d,e), so that $\pi/3$ rotations of the building block change its mechanical functionality. The resulting combinatorial metamaterial comprises an array of such blocks positioned with arbitrary orientations in a honeycomb lattice. This lattice is bipartite, see Fig. 1(d), with neighboring vertices alternating between sublattices A (blue) and B (orange). We map a deformation of a facet, indicated by an arrow in Fig. 1 to a +1 (−1) spin if it winds anticlockwise (clockwise) around an A vertex, and conversely for a B vertex. We identify ferromagnetic or antiferromagnetic interactions between neighboring spins according to their states in the building block's lowest-energy deformation, as shown in Fig. 1(d,e). These bonds are determined by the mutual winding direction of the arrows around the vertices of the honeycomb lattice: ferromagnetic (dashed line) when both displacements wind in the same direction, and antiferromagnetic (solid line) when displacements wind in opposite directions, as indicated by the circular arrows in Fig. 1(d,e).

Thus, a metamaterial specified by the orientations of all its building blocks maps to an Ising model of mixed ferromagnetic and antiferromagnetic bonds, thereby defining a bond distribution on the dual lattice. Here, the displacement arrows in Fig. 1, which sit on the facets of the hexagonal building blocks, constitute the sites of the kagome lattice, the dual of the honeycomb, and each metamaterial maps to a different bond distribution on the kagome lattice. Mechanical compatibility of a vertex in the hexagonal metamaterial is hence determined by the parity of antiferromagnetic bonds in the corresponding triangular plaquette of the kagome lattice, which can be inferred from the parity of director lines meeting at the central vertex, see Fig. 2; For an even number of antiferromagnetic bonds, as shown in Fig. 2(a), all three building blocks meeting at the vertex can simultaneously deform to their lowest-energy soft mode; If there is an odd number of antiferromagnetic bonds, as shown in Fig. 2(b), the spins are frustrated, meaning that the displacements cannot be assigned in a way that satisfies all interactions simultaneously, thus generating a topological mechanical defect, which is indicated with a black circle in Fig. 2(b).

## III. COMPATIBLE METASTRUCTURES

Lack of frustration in each plaquette implies that the entire emergent Ising model is described by what we call an even bond distribution and is thus unfrustrated, and the corresponding mechanical system is globally compatible. Note that in this system compatible configurations exhibit holographic order in the soft mode maintained by the alternating displacements of each pair of opposing facets. A global soft mode can thus be uniquely determined by the deformations along the boundary of the metamaterial; In a rhombic metamaterial consisting of $N = L \times L$ building blocks, the soft mode of a compatible architecture can be described using the $4L - 1$ principal axes running through it, see Fig. 3(a), and written in the form:

$$d_{\hat{a}}(i,j) = (-1)^{a_j + i}$$
$$d_{\hat{b}}(i,j) = (-1)^{b_i + j}$$
$$d_{\hat{c}}(i,j) = (-1)^{c_{i+j-1} + s_{ij}} \qquad (1)$$
$$s_{ij} = \begin{cases} j & i + j \leq L + 1 \\ L + 1 - i & i + j \geq L + 1 \end{cases}$$

where $d_{\hat{k}}(i,j)$ denotes the displacement along direction $\hat{k}$ of the building block in the row $i$ and column $j$, where $k = a, b, c$, and $a_j, b_i, c_\ell$ describe the deformation along the boundary, see Fig. 3(a). Hence, the number of compatible architectures $\Omega_0$ scales sub-extensively with the system size, $\ln \Omega_0 \sim \sqrt{N}$, with $N$ denoting the total number of hexagons, see Fig. 3(b). We can bound $\Omega_0$ by $2^{2L-1} \leq \Omega_0 \leq 3^{2L-1}$, see Appendix A for details. This is in contrast, for example, to the 2D combinatorial metamaterials studied in Ref. [11], in which the freedom to individually orient the constituent triangles leads to an extensive number of compatible configurations. The scarcity of such configurations in the hexagonal case highlights the importance of studying architectures beyond the compatible scope.

## IV. MECHANICAL CONSEQUENCES OF DEFECTS IN 2D METAMATERIALS

To understand defects from a global perspective, consider arbitrarily long loops of bonds in the kagome lattice. The compatibility of such loops is determined by the parity of antiferromagnetic interactions along the loop [28], which in turn, is set by the number of defects it contains, see Appendix B. For example, any loop surrounding the defect in Fig. 4(a) will consist of an odd number of antiferromagnetic interactions, whereas any loop surrounding the two defects in Fig. 4(b) will consist of an even such number. This topological characterization is related to Wilson loops, also known as holonomies of a connection, which were previously studied in the context of frustrated spin systems. The connection is defined as the product

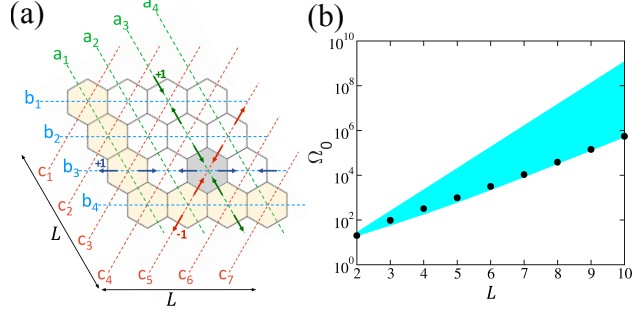

FIG. 3. The deformation field of a global soft mode is described according to holographic order and set by the deformations along the boundary, e.g., the yellow hexagons. The holographic order defines $4L - 1$ axes along which deformations alternate; $a_1 \ldots a_L, b_1 \ldots b_L, c_1 \ldots c_{2L-1}$, with $L = 4$ in this drawing. (b) The number of compatible rhombic $L \times L$ structures, exactly counted up to $L = 10$ (black dots), falls between the lower and upper bounds (blue region), and is very close to the lower bound, where the leading order is $2^{2L-1}$.

of bonds along a line; $+1$ for ferromagnetic bonds, and $-1$ for antiferromagnetic bonds. The connection along a closed loop is gauge invariant, and tells us whether there is frustration or not [27].

There is a remarkable similarity between the pattern formed by the red director lines around mechanical defects, see Fig. 4(a,b), and the point defects present in 2D nematic liquid crystals, which posses a topological charge of winding number $\pm 1/2$ [29–31]. However, the discrete orientations and positions of the building blocks in the mechanical system do not allow for a definition of a winding number, and indeed the two types of mechanical defects are indistinguishable. Locally rotating building blocks changes the number of defects by an even amount, suggesting that in our metamaterials, the parity of the defects is the topologically protected quality, see Appendix B.

We study the mechanics of the metamaterial by means of a coarse-grained model, in which we describe the complex deformation field by scalar normal displacements defined for each facet, and by assigning harmonic interactions between these scalar displacements at each hexagonal building block. The deformations of the facets serve as continuous mechanical degrees of freedom, and we can therefore write the elastic energy in the metamaterial in the following way:

$$E = \frac{1}{2} k_{ij} u_i u_j = \frac{1}{2} \mathbf{u}^{\mathbf{T}} \mathbf{K} \mathbf{u}, \qquad (2)$$

where $\mathbf{u}$ is a vector containing the displacements of all the facets in the metamaterial and $\mathbf{K}$ is a matrix containing the elastic interaction constants $k_{ij}$ between the facets $i$ and $j$. Symmetries reduce $k_{ij}$ to eight independent interaction constants $k_n$, see Fig. 5. If the arrangement of the hexagons leads to a compatible structure, the ground state of the corresponding unfrustrated Ising model describes the deformations of the global soft mode.

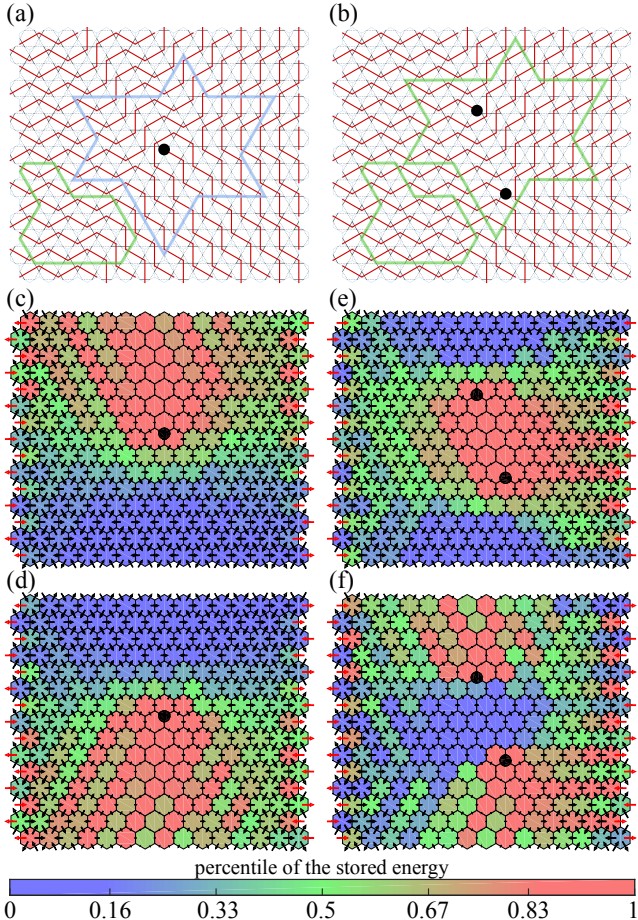

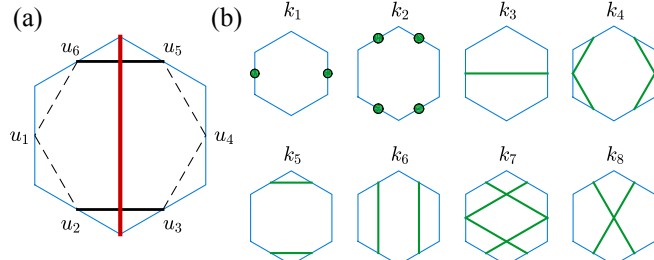

FIG. 5. (a) The coarse-grained variables $u_i$ describing the displacements of the facets. (b) For a hexagonal building block, symmetries allow eight different interaction constants between pairs of facets. The interacting facets are indicated by a connecting line, or by a circle for the diagonal terms.

over the sample, see also Ref. [26].

In realistic metamaterials, the softest deformation mode of the building block generally has finite rigidity. For simplicity, we ascribe zero energy cost to the deformation mode described in Fig. 1(b,c). This translates to the condition of a vanishing net force acting on the facets, and results in two independent equations describing the interaction constants $k_n$,

$$\begin{aligned} k_1 &= 2k_4 + 2k_7 - k_3, \\ k_2 &= k_4 - k_5 - k_6 + k_7 - k_8, \end{aligned} \tag{3}$$

see Appendix C for further details on selecting the values of the interaction constants and solving the mechanical response. Our model and calculations can be easily adjusted for finite rigidity of the softest mode, and we do not expect qualitative differences as a result.

To understand how defects can be harnessed to steer the stress distribution, note that actuating a facet of a building block defines the compatible actuation of any of its neighboring facets, given by satisfying the interaction bond between the two facets. Compatible actuation can therefore be defined along any path in the metamaterial, but can only be defined along loops containing an even number of antiferromagnetic bonds, i.e, surrounding an even number of defects.

Consider first an architecture with a single defect as portrayed in Fig. 4(a); any loop winding around it would have an odd number of antiferromagnetic interactions and thus can not be actuated compatibly. By setting compatible actuations along the opposing left and right boundaries of the metamaterial, we can control the location of the compatible and incompatible regions, thereby steering the stresses and strains to complementary parts of the system: when the actuation along the left boundary can be compatibly extended towards the actuation along the right boundary using a path below the defect, stresses concentrate above the defect, coinciding with a region of vanishing deformations, see Fig. 4(c). If we then flip the actuation of one the boundaries, so that the left and right boundaries can now be compatibly connected via a path above the defect, stresses and vanishing defor-

FIG. 4. (a) Single defect, and (b) two defects (black circles), where director lines terminate or branch and where triangular plaquettes of the kagome lattice have an odd number of antiferromagnetic bonds. Loops of interaction bonds consisting of an even (green) or odd (blue) number of antiferromagnetic bonds. (c-f) Displacement conditions at the left and right boundaries (red arrows) lead to displacements of the facets (black arrows) and to finite elastic energy stored in each building block (color-coded hexagons). The color bar indicates the percentile of the stored energy, separately calculated for each case. Single defect (c,d): Compatible actuation on each one of the boundaries concentrates the stresses (strains) at the top (bottom) or bottom (top) half of the metamaterial. Two defects (e,f): Compatible actuation on opposing boundaries concentrates the stresses either between the defects (e) or around them (f), whereas the strains concentrate in the complementary region.

However, if the system is incompatible, the lowest energy configurations of the corresponding Ising system do not necessarily describe its elastic deformations. A distinction can be made based on the different nature of the physical degrees of freedom; discrete spin degrees of freedom result in high energetic cost locally concentrated at specific (frustrated) interaction bonds, whereas continuous deformation degrees of freedom reduce the energetic cost by spreading the deviations from the local soft mode

mations concentrate below the defect, see Fig. 4(d).

In a similar manner, when the system contains multiple defects, as shown in Fig. 4(b), the regions between the defects and the boundaries can be made stressed or strained in an alternating manner, depending on the chosen compatible boundary actuation, see Fig. 4(e,f).

Note that the topological signature of a defect in our system, an odd number of antiferromagnetic interactions along a loop seemingly seeking to invert the deformation at the loop's origin, is reminiscent of the topological structure of nonorientable ribbons [32]. Therefore, it is instructive to compare their mechanical response: both systems feature a region of vanishing deformations and a region of vanishing stresses. The latter is maximally separated from the applied boundary actuations, whereas the location of the former is system-dependent. In elastic ribbons, the linear constitutive relations between stress and strain dictate that the region of vanishing deformation coincides with that of vanishing stresses. In our system, however, the local soft (floppy) mode violates these simple relations, and finite deformations persist in the region of vanishing stresses that compatibly connects the two boundaries.

## V. MECHANICAL CONSEQUENCES OF DEFECTS IN 3D METAMATERIALS

We can extend our simple approach to 3D systems, which are usually much harder to analyze. Consider the class of combinatorial metamaterials presented in Ref. [10], where cubic building blocks possess the anisotropic soft mode of deformation shown in Fig. 6(a). Similar to our 2D hexagonal metamaterials, the holographic order maintained by the building block's soft deformation mode results in sub-extensive scaling of compatible architectures with system size, see Appendix D. Here too, we define ferromagnetic and antiferromagnetic bonds between adjacent arrows describing deformations in the building block, according to whether or not they maintain the same winding direction around the shared lattice edge between them, as depicted by the dashed and solid lines in Fig. 6(b). Again, compatibility is associated with parity of antiferromagnetic interactions along closed loops. However, simple connectedness is removed by point defects in 2D, but by line defects in 3D. This has well known consequences in materials: for instance, dislocations are point defects in 2D, but line defects in 3D. Similarly, incompatibilities are described as line defects in this 3D system while they are point defects in the 2D system [30], see Fig. 2(b). In 2D, our elementary loops on the dual lattice wind around lattice vertices. In 3D, they wind around the shared edge of four cubes, see Fig. 6(c,d), which is identified as a defect if the number of antiferromagnetic bonds surrounding it is odd, as shown in Fig. 6(d).

Because the parity of antiferromagnetic interactions along a 3D loop must remain unchanged as it morphs be-

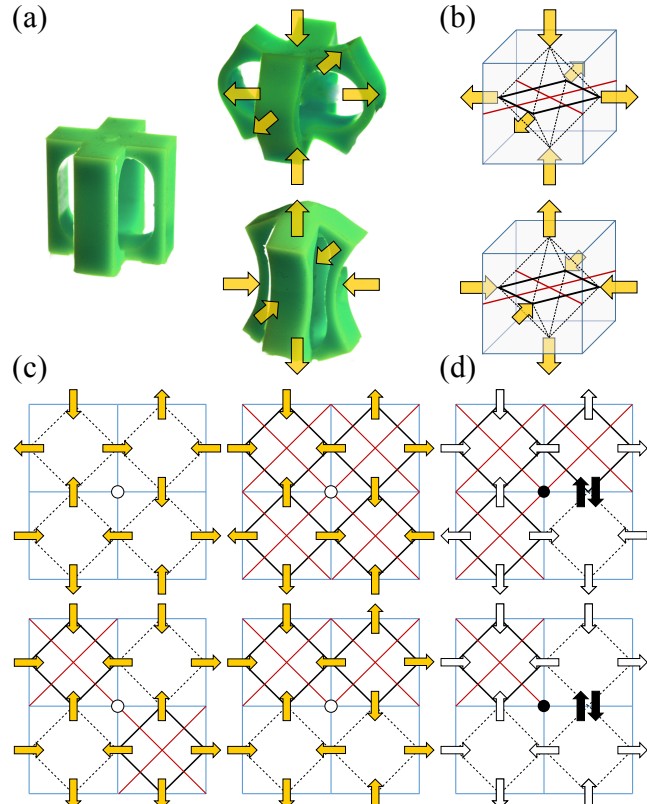

FIG. 6. (a) 3D cubic building block and its soft deformation mode (reproduced from Ref. [10]). (b) Deformation sign between two adjacent facets is preserved (flipped) if deformations wind in the same (opposite) direction with respect to the common edge between them. Ferromagnetic sign-preserving (antiferromagnetic sign-flipping) interactions are indicated by dashed (solid) lines connecting the two facets. A red cross drawn perpendicular to the sign-flipping interactions designates the orientation of the building block. (c) Top view of a compatible and (d) an incompatible edge consisting of an even (odd) number of antiferromagnetic interactions, as indicated by white (black) circle. The number of antiferromagnetic interactions can be inferred from the parity of red lines meeting at the central edge.

tween the facets and over the non-frustrated lattice edges of an even bond distribution, defected edges must join to form defect lines. These must either close into loops, or extend between the boundaries of the system [33]. Starting from a compatible configuration and rotating a single building block leads to two parallel loops of frustrated edges. In that sense, mechanical defects in 3D are reminiscent of the topologically neutral disclination loops seen in 3D active nematics [34].

To study the mechanics of the system, we imply the coarse-grain model described in Eq. (2) to the metamaterial comprised of the cubic building blocks described in Fig. 6(a). We can identify the facets of the cubic building blocks with those of the hexagonal building block, and use Eq. (3) together with $k_4 = k_7$ and $k_5 = k_6$ to describe the interaction constants (a total of six in-

dependent interaction constants). Textured actuation along the boundary of the metamaterial can steer strains and stresses around the complex lines of frustrated edges in different fashions, giving rise to different mechanical functionalities for a given structure.

Consider first the simple extension from 2D to 3D, namely frustrated edges connected to form a straight defect line terminating on opposing faces of the metacube. The metacube can be compatibly actuated on the opposing defect-free faces (parallel to the $(y, z)$ plane) in such a way that stresses can be steered around the defect line, and can be localized on the one half of the material, whereas the strains are larger in the other half, cf. Fig. 7(a). While this scenario is reminiscent of 2D stress steering, the structure's extra dimension offers a richer plethora of possibilities. For instance, by actuating the same metacube through its incompatible faces (those parallel to the $(x, z)$ plane), we can generate more complex response patterns such as a twisted stressed region, as shown in Fig. 7(b). In this case, since we cannot force the entire $(x, z)$ faces in a compatible manner, we introduce a cut running from the location of the defect to the system's boundary, and do not actuate along this cut. When the remaining face is actuated in a compatible manner, stresses concentrate along the designated cut. We set the cuts on two opposing faces to be orthogonal to one other, thus causing a 3D twist in the stress concentration inside the metamaterial, cf. Fig. 7(b). The other fundamental defect topology we consider is a closed defect loop, see Fig. 7(c,d). Here, by compatibly actuating opposing facets parallel to the $(x, z)$ plane, we can concentrate the stresses outside or inside the loop.

Finally, in complex topologies featuring multiple defect lines, such as the defect cross arrangement presented in Fig. 7(e), stress and strain concentrate in complementary regions that alternate around the defect lines with respect to the boundary conditions. Therefore, by compatibly actuating the facets opposing the defect cross (parallel to the $(x, z)$ plane), we can concentrate the stresses in two separate quadrants. Note that taking cross sections of the stress concentration maps through planes rotated around the $y$ axis results in images reminiscent in nature to Fig. 4(e,f). Also note that our combinatorial approach allows us to generate these different defect patterns both with periodic and with non-periodic structures, cf. Fig. 7(f), however, the described features of the mechanical response remain unchanged, as shown in Appendix E.

## VI. DISCUSSION

The framework we present maps the soft modes of deformable building blocks to ferromagnetic and antiferromagnetic interactions on the underlying dual lattice of the metamaterial that is formed by these blocks. The orientations of all blocks in the structure define a bond distribution on this lattice, and that, in turn dic-

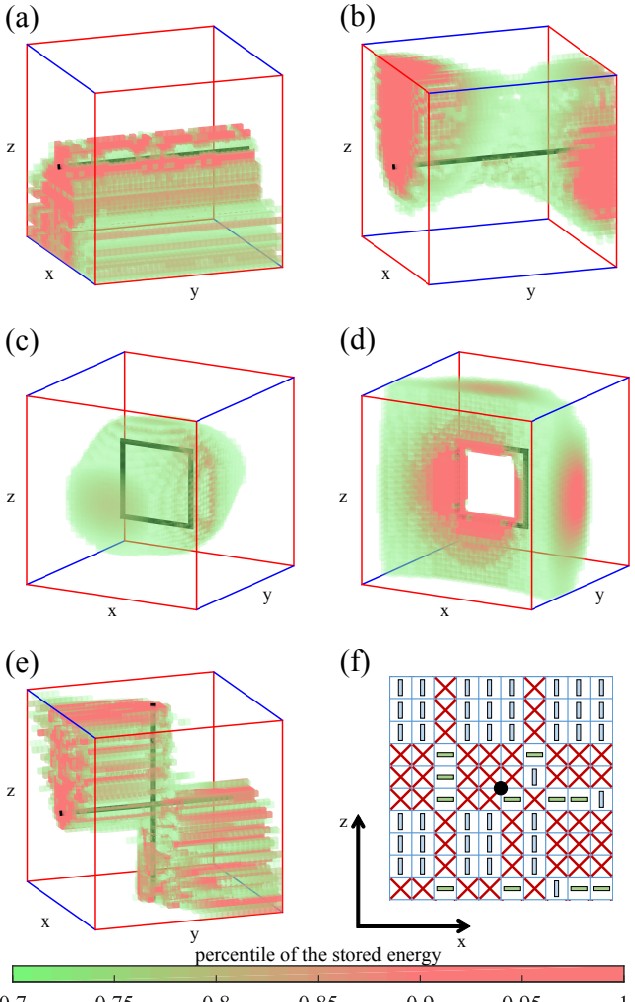

FIG. 7. (a) Compatible actuation on the back and front faces can concentrate stresses beneath the defect line. (b) Twisting the stressed region through actuation on the incompatible left and right faces. (c,d) Compatible actuation on the front and back faces concentrates stresses outside (c) or inside (d) a defect loop. (e) Compatible actuation on the front and back faces concentrates stresses on two separated quadrants. Color bar indicates the percentile of the stored energy, separately calculated for each case. The faces on which the actuation is applied are indicated by red frames. (f) Partial cross section close to the centered defect line of the structures in (a,b), showing the non-periodic internal architecture. In this top view, building blocks oriented along the $y$ axis are represented by a red cross (cf. Fig. 6(c,d)), whereas building blocks oriented along the $x$ and $z$ axes are represented by horizontal and vertical rectangles, respectively. All calculations are for metacubes of dimension $35 \times 35 \times 35$.

tates the compatibility, frustration, and topological defects of the combinatorial metamaterial. We provide detailed demonstrations for such combinatorial metamaterials constructed of two specific hexagonal and cubic building blocks. However, our framework is suitable for many types of metamaterials made of deformable blocks

with arbitrary internal interaction rules. It also provides a platform to describe metamaterials with vacancies, or constructed by mixing different types of building blocks. Our approach enables programming metamaterials with complex defect patterns, as well as devising spatially textured actuations that yield different mechanical functionalities from a single sample. Controlling and steering the mechanical response in the bulk of 3D metamaterials could enable adaptive failure control, could potentially be implemented in nematic elastomers [35], and may also lead to additional applications such as steering waves [36], or to drive active matter [37–39].

## ACKNOWLEDGMENTS

We thank Corentin Coulais, Martin van Hecke, Roni Ilan, Yoav Lahini, Ron Lifshitz, Anne S. Meeussen, Carl Merrigan, Ivan Smalyukh, and Eial Teomy for fruitful discussions. This research was supported in part by the Israel Science Foundation Grants No. 968/16 and 1899/20, by the Israeli Ministry of Science and Technology, and by the National Science Foundation Grant No. NSF PHY-1748958. Y.S. thanks the Center for Nonlinear Studies at Los Alamos National Laboratory for its hospitality. The work of C.N. was carried out under the auspices of the U.S. DoE through the Los Alamos National Laboratory, operated by Triad National Security, LLC (Contract No. 892333218NCA000001).

## Appendix A: Compatible 2D structures

The number of compatible 3D metacube configurations has been investigated in Ref. [10]. We will complete the discussion of counting compatible configurations by similarly providing lower and upper bounds for our 2D hexagonal metamaterials. Below, we discuss non-periodic 3D metacube structures, and will also show that a tighter lower bound can be derived for the number of compatible metacubes, compared to the lower bound presented in Ref. [10]. Note that the following discussion concerning compatible configurations and global floppy modes also applies to identifying the softest global modes, in case that the mode described in Fig. 1(a) is the softest mode, but not necessarily completely floppy.

Counting the compatible configurations is equivalent to identifying all the global floppy modes in the system, as the displacement field of a global floppy mode uniquely defines the constituent architecture. The floppy mode of an individual hexagon exhibits alternating displacement directions between each pair of opposing facets. In a compatible structure, all building blocks can deform simultaneously according to their floppy mode, and thus the displacement field of a global floppy mode maintains holographic order in the form of alternating displacements along any direction into the metamaterial. Because of that, such displacement fields in an $N = L \times L$ rhombic

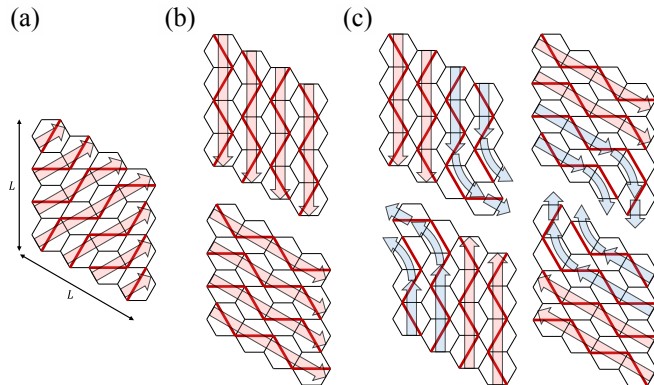

FIG. 8. (a) $2L - 1$ straight lines, along the diagonal principal directions of the honeycomb lattice. (b) $L$ straight lines, along the horizontal and vertical principal directions of the honeycomb lattice. (c) Straight vertical or horizontal lines, followed by a single curve. Given an $L \times L$ structure, the number of straight lines in (c) can range between 0 and $L - 2$. In (a), (b) and (c) red arrows represent two possibilities for a straight zig-zag line, whereas blue arrows represent a line uniquely determined by the curve.

metamaterial can be described by the boundary displacements along the $4L - 1$ principal axes running through it, see Fig. 3 and Eq. (1). Therefore, there are up to $2^{4L-1}$ candidates for the global displacement fields, which correspond to up to $2^{4L-2}$ different global floppy modes, and similarly compatible structures.

However, some of these candidates for global floppy modes will result in building blocks, in which all the displacements point outwards, or all inwards, violating the local floppy mode of the individual building blocks. Some of these unwanted modes can be avoided by considering that at least along the $(2L - 1)$-long boundary, the displacement field at each building block is consistent with a floppy mode of one of the orientations, see yellow hexagons in Fig. 3(a). This can be easily verified by noting that choosing the orientation of each building block along the boundary is sufficient to define the displacement field (up to global reversal), and thus to define potential global floppy modes. We therefore arrive at an upper bound of $3^{2L-1}$ for the number of $L \times L$ compatible configurations as there are three possible orientations per hexagon.

A lower bound for $\Omega_0$, the number of compatible architectures, is obtained by presenting systematic strategies to design such structures. Consider the red lines depicted in Fig. 1(d,e), which designate the orientation of the building blocks. In a compatible structure, these red indicators connect to form a pattern of zig-zag lines that must not terminate or bifurcate, see Fig. 2. We can therefore consider a simple strategy to design compatible structures, in which the zig-zag lines run along the $2L - 1$ parallel $c_i$-axes or along the $L$ $a_i$- or $b_i$-axes of the parallelogram, as shown in Fig. 8(a,b). Along each such axis, there are two possibilities for the zig-zag pat-

tern, as it can be mirrored with respect to the axis whilst still keeping the same path, see red arrows in Fig. 8. There are therefore $2^{2L-1}$ structures constructed by zigzag lines confined to the $c_i$-axes and $2^L$ structures with zig-zag lines along the $a_i$- or $b_i$-axes. We therefore arrive at $\Omega_0 \geq 2^{2L-1} + 2^{L+1}$.

By adding configurations in which only some of the zig-zag lines stay along the principal axes, whilst the others curve and are thus restricted to a specific pattern, see Fig. 8(c), it could be easily verified that $\Omega_0 \geq 2^{2L-1} + 2^{L+2} - 4$, for $L \geq 2$. Additional contributions with multiplicity that scales as $2^L$ can be obtained by considering more complex patterns, yet $2^{2L-1}$ remains the leading term in the large $L$ limit.

Finally, the exact number of compatible configurations was calculated for up to $10 \times 10$ systems by manually considering all the deformation fields that obey holographic order, yet do not violate the floppy mode in any of the building blocks. The exact count follows very closely the provided lower bound, see Fig. 3(b).

## Appendix B: Topological defects

The fundamental property of topological defects is that their removal requires tampering with the system at arbitrarily great distances away from the defect itself [40]. In our system, we define a mechanical defect as an interface between neighboring building blocks that induces an odd number of antiferromagnetic bonds at the corresponding elementary loop on the dual lattice, see Fig. 1. In this appendix, we demonstrate that it is the parity of such defects which determines the far-away topological implications, and resultantly, that only an odd number of such mechanical defects constitute a topological defect.

The signature of defects is observed through the parity of antiferromagnetic bonds along loops on the bond distribution, which in turn alludes to mechanical compatibility. A space containing no mechanical defects, and therefore only elementary loops with an even number of antiferromagnetic bonds, induces an even bond distribution [28]. Over such a space, any loop homotopic to an elementary loop also consists of an even number of antiferromagnetic bonds. In fact, it can be generally argued that over the space of an even bond distribution, homotopic loops share the same parity of antiferromagnetic bonds. Consider the loops depicted in Fig. 9(a). The solid loop is homotopic to the loop in which the path between points $\mathcal{A}$ and $\mathcal{B}$ is replaced by the dashed path. To prove that both loops share the same parity of antiferromagnetic bonds, we observe the loop that is formed by the solid and dashed paths connecting points $\mathcal{A}$ and $\mathcal{B}$. Note that this loop is homotopic to an elementary loop in the even bond distribution, and therefore consists of an even number of antiferromagnetic bonds. As a result, both paths share the same parity of antiferromagnetic bonds, and one can be replaced by the other without changing the overall parity. However, loops whose ho-

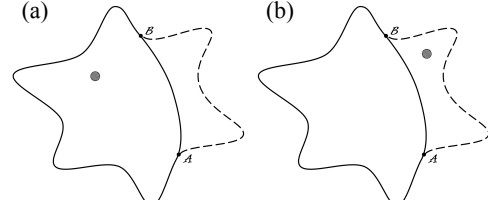

FIG. 9. Parity of antiferromagnetic bonds along homotopic loops. The dashed circles represent defects - elementary loops with an odd number of antiferromagnetic bonds. These circles constitute holes in the space of the even bond distribution.

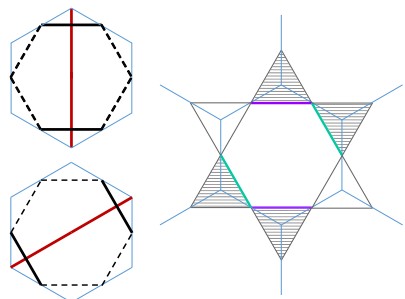

FIG. 10. Hexagonal building block before (top left) and after (bottom left) a $\pi/3$ rotation. Induced changes to the bond distribution (right). Violet and green lines represent antiferromagnetic bonds that were replaced by ferromagnetic bonds, and vice versa. The elementary loops surrounding the rotated building block that changed their parity due to the rotation are indicated by dashed background.

motopy requires crossing over a defect, have a different parity of antiferromagnetic bonds. Consider the loops depicted in Fig. 9(b). The solid loop is no longer homotopic to the loop in which the path between points $\mathcal{A}$ and $\mathcal{B}$ is replaced by the dashed path. In this case, the loop that is formed by the solid and dashed paths connecting points $\mathcal{A}$ and $\mathcal{B}$ is homotopic to the elementary loop surrounding the defect, and hence consists of an odd number of antiferromagnetic bonds. Replacing the solid path with the dashed path therefore changes the parity of antiferromagnetic bonds.

Switching the parity of the total number of defects, therefore, requires changing the bond distribution infinitely far away (or, equivalently, all the way to the boundary of the system). We have therefore proven that locally rotating building blocks could only alter the number of defects by an even amount, and that the parity of the defects is topologically stable, which means that the defect charge has $\mathbb{Z}_2$ symmetry. To provide insights specific to our system, it is instructive to observe the effects on the bond distribution of rotating a single building block. Figure 10 demonstrates that such a local change to the configuration changes an even number of bonds, which can only alter the number of defects by an even amount. This result is independent of the specific shape of the soft mode (and hence, the inner bond distribution)

of our building blocks. In fact, we can exchange the type of the building block altogether and still observe a total even number of changed bonds. This argument holds because the inner bond distribution, derived from the local shape of the soft mode, by construction must consist of an even number of antiferromagnetic bonds.

In 3D, where the space is embedded with defect lines, the homotopic properties of loops and parity of antiferromagnetic bonds can be inferred from the three projections of the loop into planar loops, together with the corresponding projections of each segment of the defect lines into defect points in the perpendicular plane. Combining the number of defect points inside the three planar loops gives the equivalent of the winding number of the 3D loop around the defect lines.

## Appendix C: Mechanical response model

To understand how to choose the interaction constants $k_{ij}$ in Eq. (2), it is instructive to observe a single building block, where $\mathbf{K}_s$ is a 6×6 matrix containing the elastic interaction constants, both for the 2D hexagons and for the 3D cubes. From symmetry considerations, it can be easily seen that both the 2D hexagonal and the 3D cubic building blocks have only two types of facets, two along the minority axis and four along the majority axes. It can also be easily verified that there are eight (six) possible different interaction constants $k_{ij}$ for the hexagonal (cubic) building block, see Fig. 5. These interaction constants take positive (negative) values if the energy decreases when the facets displace oppositely (similarly) with respect to the building block.

Satisfying Eq. (3), i.e, a vanishing net force on the facets when deformed according to the desired floppy mode, guarantees that this mode indeed costs no energy, and that it is an eigenmode of the matrix $\mathbf{K}_s$ with a zero eigenvalue. Finally, in order for the building block to be mechanically stable, the remaining interaction constants were chosen such as that the other eigenvalues of $\mathbf{K}_s$ are all positive. In our numerical demonstrations of the mechanical response in the presence of defects we used the arbitrary values $k_1 = 0.5$, $k_2 = 0.5$, $k_3 = -0.289$, $k_4 = 0.065$, $k_5 = -0.219$, $k_6 = -0.027$, $k_7 = 0.041$, $k_8 = -0.149$ in 2D, and $k_1 = 1$, $k_2 = 2$, $k_3 = 0.246$, $k_4 = k_7 = 0.311$, $k_5 = k_6 = -0.929$, $k_8 = 0.48$ in 3D. We also tested other sets of values and observed no qualitative difference in the results. It should be noted that it is possible to adjust the $k_i$ selection for finite rigidity by setting the desired soft mode to be the eigenmode of the matrix $\mathbf{K}_s$ with the lowest eigenvalue, however we do not expect qualitative changes as a result of switching from a floppy mode to a soft mode.

In order to find the mechanical response of a metamaterial structure to a set of externally applied constraints on some of its facets, we find the deformation field such that the net forces on the remaining free facets vanish. Since we assume a harmonic energy term, the forces are linear in the deformations and a set of linear equations $\mathbf{K}_f \mathbf{u}_f = \mathbf{b}$ can be written and easily solved numerically, where $\mathbf{K}_f$ is the matrix describing the interaction between the free facets $\mathbf{u}_f$, and $\mathbf{b}$ is a set of the external forces applied on these facets. Solving this equations set requires inverting matrix $\mathbf{K}_f$, which scales as $L^2 \times L^2$ for $L \times L$ hexagonal metamaterials and as $L^3 \times L^3$ for $L \times L \times L$ cubic metamaterials. Note that because the energy landscape is a convex second-order expression of the deformations, the thereby found extremal deformation field is guaranteed to be an energy minimum.

## Appendix D: Compatible 3D structures

We construct compatible $L \times L \times L$ metacubes by carefully stacking $L \times L \times 1$ layers. First, we observe the conditions under which such layers are individually compatible. We refer to building blocks whose minority axis is oriented in the $i$ direction as $i$ blocks, and to lattice edges along the $i$ direction as $i$ edges. To satisfy compatibility in the $(x, y)$ plane, $z$ blocks must reside in a pattern of alternating regions demarcated by a set of vertical and horizontal lines that form a subset of all grid lines, see Fig. 11. This guarantees that there are 0, 2 or 4 red lines meeting at each $z$-edge. There are thus $\left(2^{L-1}\right)^2$ possible ways to select the subset of the vertical and horizontal lines.

A given selection of the vertical and horizontal lines, as demonstrated by solid black lines in Fig. 11, defines two possible colorings for the $z$ blocks and complemen-

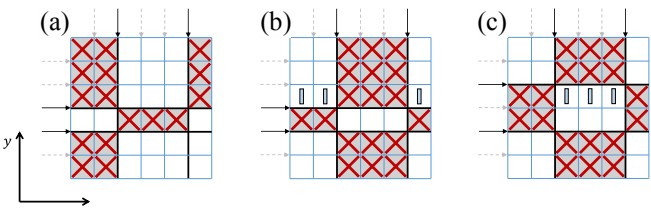

FIG. 11. Compatible layers: $z$ blocks, indicated by a red cross, cf. Fig. 6, are bound to alternating regions between a subset of horizontal and vertical grid lines, indicated by solid black lines. The empty blocks in the complimentary regions are free to choose between $x$ and $y$. The horizontal and vertical lines are each selected from a set of $\{L - 1\}$ possible grid lines. The chosen (unchosen) lines are indicated by solid black (dashed gray) arrows. (a) and (b) depict the two possible colorings for the same selection of vertical and horizontal lines. If there are $f$ free blocks in coloring (a), then there are $L^2 - f$ free blocks in coloring (b). (c) Row swapping with respect to the layer at (b). Every block in the $z$ direction was replaced with a block in the $y$ direction and vice versa. $y$ blocks are indicated by elongated rectangles in the $y$ direction. The swapping changes the selection status of the horizontal lines bounding the line; from a solid arrow indicating a selected line to a dashed arrow indicating an unchosen line, and vice versa.

tary regions, in which the complementary regions consist of a total of $f$ or $\left(L^2 - f\right)$ blocks, see Fig. 11(a,b). Each individual block in the complementary regions is free to choose between being an $x$ block or a $y$ block. There are thus $2^f$ or $2^{L^2-f}$ different ways in which the orientations of the blocks in the complementary regions can be chosen. Note that $2^f + 2^{L^2-f} \geq 2^{L^2/2+1}$, and therefore the number of compatible $L \times L \times 1$ layers satisfies

$$\Omega_L \geq 2^{L^2/2+2L-1}. \tag{D1}$$

The second stage of our procedure involves stacking compatible layers. Compatible stacking requires an even number of red lines meeting at the $x$ and $y$ edges between the layers, in addition to the $z$ edges inside each layer. $x$ or $y$ edges may receive 0, 1 or 2 red line contributions from each layer, depending on how many blocks are facing the $x$ or $y$ directions on either sides of the edge. Note that stacking the layer on itself always results in a compatible interface as this number of red lines doubles.

Consider a special case, in which along a row in the $x$ direction, all the free blocks were chosen to be in the $y$ direction, see Fig. 11(b). Now consider a stacking where in the next layer along the same row, every block in the $z$ direction was replaced with a block in the $y$ direction and vice versa, see Fig. 11(c). This change does not change the number of red lines on any $x$ edge between the layers compared to stacking the layer on itself. The $y$ edges between the original row and the swapped row will have exactly 2 red lines and will therefore also be compatible. Finally, the described swapping is equivalent to changing the selection status of the horizontal lines bounding the described row, see Fig. 11(c), which means that the swapped layer will still satisfy compatibility on all its $z$ edges. Therefore, when such rows exist, they can be swapped freely between the layers without compromising the stacking compatibility.

The described stacking process can easily be used to create non-periodic compatible structures. Consider a compatible layer comprised only of $z$ and $y$ blocks, where only vertical lines were chosen to separate between $z$ and $y$ regions. There are $2^{L-1}$ possibilities to select the vertical lines for this reference layer. However, each row in this layer can be swapped, allowing $2^L$ compatible stacking possibilities. The number of compatible $L \times L \times L$ metacubes that can be constructed in this way is a lower bound to the total number of compatible metacubes

$$\Omega_0 \geq 3 \cdot 2^{L^2+L}, \tag{D2}$$

where a factor of 2 was included to account for column swapping as well as row swapping, and a factor of 3 was included to account for stacking planes in the $x$ and $y$ directions, equivalent to 6 rotations in space. Note that the exact same lower bound was found in Ref. [10], using different arguments.

However, our approach for non-periodic stacking easily allows us to tighten this lower bound by also considering $\Omega_{xyz}$, the number of structures created from reference

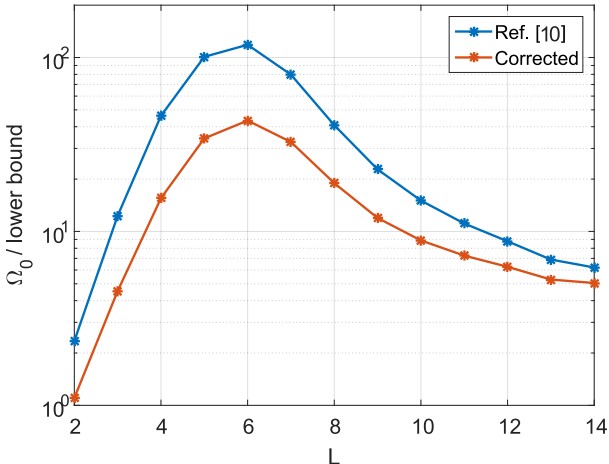

FIG. 12. Ratio between the exact number of compatible metacube structures and the lower bounds, using the original lower bound described in Ref. [10] (blue) and the tighter lower bound of Eq. (D4) (red).

layers with rows containing also $x$ blocks. When creating structures from such layers, the aforementioned rows cannot be swapped between the stacked layers. Note that unlike the structures described for the lower bound in Ref. [10] or in the main text, $\Omega_{xyz}$ structures contain blocks of all three possible orientations. A layer with $1 \leq k \leq L$ rows containing $x$ blocks has $2^{L-k}$ compatible stacking possibilities. Consider one of the $2^{L-1}$ possible choices of the vertical lines, the two coloring of which define $a$ or $(L-a)$ non-$z$ blocks along each row. To avoid double counting, within the chosen $k$ rows, at least one of these blocks must be an $x$ block while the rest can choose between $x$ and $y$ blocks. We can then use the inequality $\left(2^a - 1\right)^k + \left(2^{L-a} - 1\right)^k \geq 2\left(2^{L/2} - 1\right)^k$ to arrive at

$$\begin{aligned} \Omega_{xyz} &\geq \sum_{k=1}^{L} 3 \cdot 2^L \binom{L}{k}\left[2\left(2^{L/2} - 1\right)^k\right] 2^{(L-k)L} \\ &= 3 \cdot 2^{L^2+L+1}\left[\left(1 + 2^{-L/2} - 2^{-L}\right)^L - 1\right], \\ \Omega_{xyz} &\geq 3L \cdot 2^{L+1} \cdot \left(2^{L^2/2} - 1\right), \end{aligned} \tag{D3}$$

where a factor of 6 was included to account for structure rotations in space, and at the last step only the leading term of a Taylor expansion was kept. Finally, we arrive at

$$\Omega_0 \geq 3 \cdot 2^{L^2+L} + 3L \cdot 2^{L+1} \cdot \left(2^{L^2/2} - 1\right). \tag{D4}$$

Figure 12 shows the improvement in the lower bound as a result of the added term.

## Appendix E: Non-periodic incompatible 3D structures and their mechanical response

Straight defect lines can be achieved by stacking layers containing defects in the desired locations, see Fig. 13. By implementing the swapping rule discussed earlier, no additional defects are created within or between the stacked layers, and the defects inside the different layers connect to form a continues line. This way, we can easily design structures with multiple parallel defect lines at designated locations.

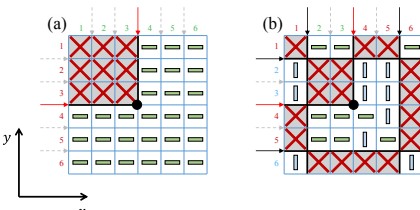

FIG. 13. A defect is formed by truncating together a vertical and a horizontal line, indicated by the red arrows. $x$ ($y$) blocks are indicated by green (blue) elongated rectangles. Swappable rows (columns) are numbered in blue (green). (a) A periodic design is constructed by self stacking the presented layer. A non-periodic design can be constructed by swapping any of the columns in (a) or by swapping of permitted columns or rows in (b). Note that switching from columns swapping to row swapping requires going through the presented reference layer.

To construct a structure with a defect loop, we devised two layers such that the interface between them will result in a 2D defect loop, see Fig. 14. This way we can design non-periodic structures with 2D defect loop of an arbitrary shape. Note that in a similar fashion we can also design multiple arbitrary loops on parallel planes.

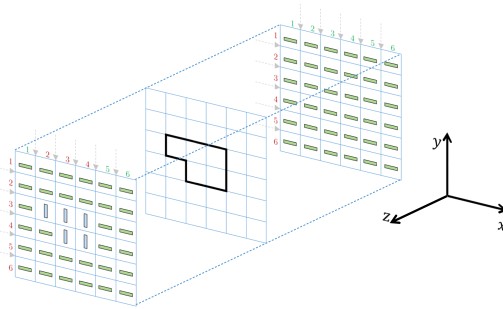

FIG. 14. To create a defect loop, a layer with $x$ blocks only is stacked on top a similar layer that also features an enclosed region of $y$ blocks. Separately, each of these layers is compatible. However, when stacked, a defect loop matching the contour of the $y$ blocks is formed between them. This is because along this contour each $x$ edge ($y$ edge) receives 3 (1) red line contributions. Without compromising the shape and position of the defect loop, all the columns outside the cross section of the loop can be swapped in the front layer, as well as all the columns in the back layer.

To construct a defect cross we need to control the location of two defect lines that are perpendicular to one another. We create such defect lines using transitions between three reference layers, see Fig. 15. If the first and third layers are stacked directly on top of one another, the two perpendicular defect lines are formed in the same plane, resulting in a defect cross.

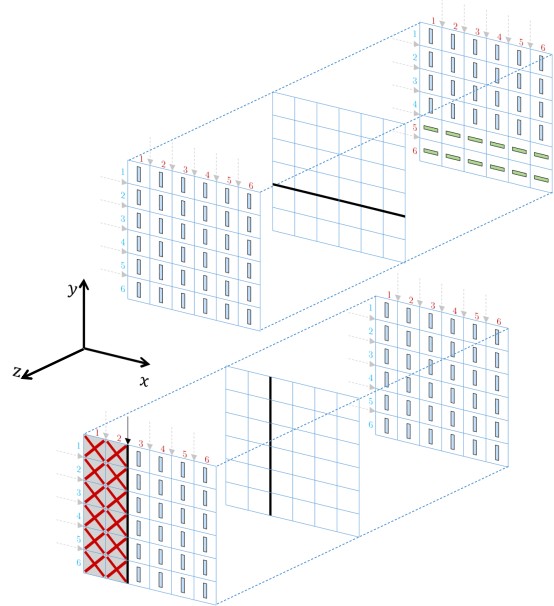

FIG. 15. Defect cross: The first reference layer (top right) contains two vertical regions of $x$ and $y$ blocks, the second layer contains only $y$ blocks, and the third (bottom left) contains two horizontal regions of $y$ and $z$ blocks. In the transition between the first (second) and second (third) layers, along the boundary between the vertical (horizontal) regions, the $x$ ($y$) edges receive one (three) red line contribution and hence a defect line is formed parallel to the $x$ ($y$) direction. The rows in the top region before the first transition, as well as all the rows after the second transition can be swapped without changing the resulting defect locations.

We presented various ways in which different structures can be designed with the same underlying defect pattern. These included self stacking of layers, as well as swapping of rows and columns. Here, we compare the mechanical response of two structures with a straight defect line; a periodic and a non-periodic structure, see Fig. 16. Even though the exact spatial distribution of stresses varies between the periodic and non-periodic structures, the ability to steer stresses around the defect lines is qualitatively similar. Note that the textured boundary condition applied to the faces of the structure depends on the internal architecture and thus differs completely between the two cases.

(a)  (b)

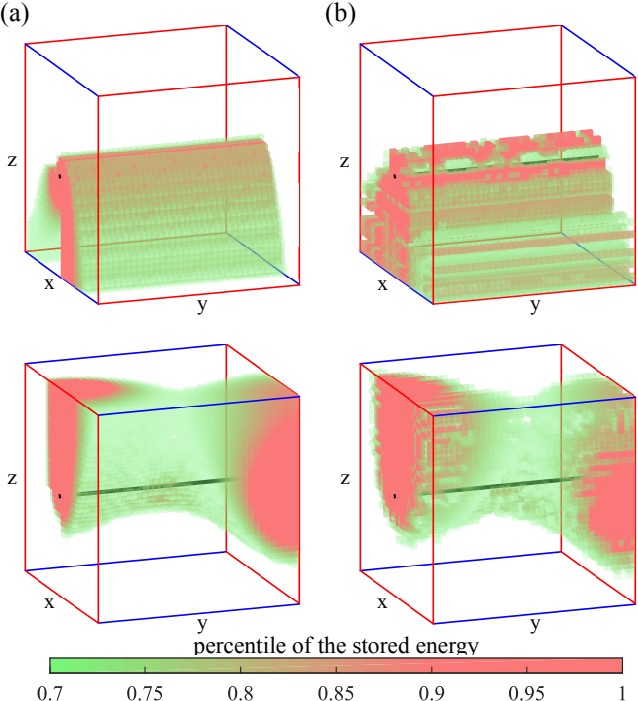

percentile of the stored energy

0.7    0.75    0.8    0.85    0.9    0.95    1

FIG. 16. Mechanical response of periodic (a) and non-periodic (b) metamaterials with the same defect structure. In (a), a simple reference layer, similar to the one presented in Fig. 13(a) is self stacked along the $y$ direction. In (b), a complex reference layer, similar to the one depicted in Fig. 13(b), is stacked along the $y$ direction with multiple rows or columns swaps between consecutive layers. Top (bottom) - applying a textured boundary condition to the $(y, z)$ faces $((x, z)$ faces) in order to steer the stresses below the defect line (twist the stressed region).

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
