# Peer review of "Putting a spin on metamaterials: Mechanical incompatibility as magnetic frustration"

_SciPost Physics_

## Round 2 · Referee Report · Anonymous (Referee 1) · 2021-3-9

Strengths
- The framework presented in this work is general and promising.
Weaknesses
- Very descriptive paper with a lack of concrete calculations or data to back up their claims. Most quantitative results are to be found in the appendix.
- Some key concepts are not discussed enough or need clarification.
Report
In this paper, the authors exploit an analogy between frustrated spin systems and the mechanical deformations of metamaterials to shed light on the behavior of the topological defects, and the role they play in the mechanical response of the material. They map the deformations of a metamaterial with a classical system made of Ising spins with FM and AF interactions, resulting on a local topological constraint reminiscent of spin-ice systems (although with a zero-point configurational entropy which is not extensive). Frustration in the spin representation is associated to mechanical incompatibility, providing a useful framework to study deformations of metamaterials. I find the overall work interesting, well presented and worth publication. However, there are several aspects that I think should be clarified before publication.
-
The authors should explain better how to control the location of the defects by actuating the metamaterial from the boundaries. I believe this is a key point of the paper which remains somehow loose in the main text. The authors could explain more concretely how the material reacts to a perturbation and analyse in more depth a precise situation. Besides, much of the connection between defects and mechanical response is to be found in the appendix B. The latter is not appropriately mentioned in the main text: the appendix is only cited as a place where to look for details of a coarse-grained model, but, to me, it sheds light on the mechanisms controlling the response of the system. I found the results presented in the main text, basically the definition of the mapping deformation<->spins, quite slim and not enough to understand that 'By actuating the metamaterial along two opposing boundaries... we can control the location of the incompatible region'. I overall find quite unpleasant to go back and forth to the appendix to find the information I need to understand the paper.
-
Related to the latter point: a study of the dynamics of topological defects when actuating the system would be useful to understand the response of the system from the viewpoint of defects. This is a general criticism, but I think the paper is mostly focused on the description of the mapping but does not provide much quantitative results to sustain their claims. At this level, the paper is largely descriptive, and it is only when looking at the appendix that one realizes that some calculations have been carried.
-
The spin assignment could be clarified further. For instance showing the two sublattices A and B. I found hard to follow how the mapping between a deformation configuration and a spin works in practice, and it is important to be very clear about that from the very beginning, since the rest of the work is based on it.

Anonymous on 2021-02-02 [id 1198]
What was hard to believe being possible just a decade ago (and with old textbooks saying that this was in fact impossible), is now realized in artificial materials called metamaterials, a class of man-made materials where properties are pre-designed to be very different from those in their natural/classical counterparts. This article describes such mechanical metamaterials. It also defines different types of topological defects, again extending/redefining the concept well beyond the classic one. What is more, however, is that the new effects arising from mechanical incompatibility can be paralleled with and mapped onto the behavior of more standard magnetic systems with intrinsic frustration. This theoretical work is inspired recent experimental realization of mechanical metamaterials, but it also describes new avenues for realizing such unusual materials in systems like liquid crystal elastomers, potentially also combined with director patterning and spatially resolved polymerization (e.g. based on two-photon polymerization could be one way to go). The paper is well written, the schematics and graphs are clear. I can see this work potentially opening the doors to the realizations of various unusual 3D spin ice systems for fundamental research (and they will join all kinds of colloidal, skyrmionic and many other model systems of spin ice pursued recently!), as well as, potentially, pre-programmed mechanical actuators and artificial muscles with complex mechanical responses defined by topological defect networks within the meta materials, as well as through engineering their responses to external stimuli. I have seen some preliminary results of this work at a KITP workshop in Santa Barbara - nice to see that this research is now developed to a very complete and in-depth study that, hopefully, will inspire more developments

---

## Round 2 · Referee Report · Anonymous (Referee 2) · 2021-4-9

Strengths
Weaknesses
Report
Requested changes
1) The manuscript alludes to Wilson loops. While this is a well-known object, it would be useful to remind the reader of their definition. More importantly, this would give an opportunity to explain how these show up in the context of mechanical metamaterials, how they are computed, etc. (Usually, Wilson loops are holonomies of a connection. What is the connection here? etc.)
2) Consider the sentence "Note that in this system compatible configurations exhibit holographic order in the soft mode maintained by the displacements of each pair of opposing facets." This is a very captivating sentence, but the physics behind is not completely clear. How is the "holographic order" defined? Why is it holographic? What is the basis for these statements?
3) The discussion on mechanical defects is also very interesting. It is not very clear to me how the defects are controlled from the boundary in Fig. 2. A more detailed discussion of the control parameters and what control they allow would be welcome. I suggest to show explicitly the statement: "Locally rotating building blocks changes the number of defects by an even amount, suggesting that in our metamaterials the parity of the defects is the topologically protected quality".
On this point, I would like to ask: shouldn't it still be possible to write down a topological charge, that would have value in Z_2? If not, what prevents one from doing that? Defects with Z_2 charges can indeed occur. This is the case in 3D nematics or superfluid 3-He, in which \pi_1 is Z_2 (see for instance (5.12) in V.B.3 in Mermin's review on the topological theory of defects). The review of Alexander, Chen, Matsumoto, and Kamien in RMP gives a nice picture of how to transform a +1/2 disclination line into a -1/2 disclination line in 3D (see Fig. 4). Can the authors harness ideas from these sources to make their point more precise?
4) The statement: "If the arrangement of the hexagons leads to a compatible structure, the ground state of the corresponding unfrustrated Ising model describes the deformations of the soft mode. However, if the system is incompatible, the lowest energy configurations of the corresponding Ising system do not necessarily describe its elastic deformations." is intriguing. It would be very nice to have an explicit map written down, so that the readers can see for themselves when it works and when it fails from the equations.
5) Minor issues: - Fig. 2: please add "percentile of the stored energy" below the color bar - I don't understand the use of italic in the paper. - What is the meaning of "-" in "(y, z)− plane"? - Fig. 10: typo "Ref. [1]" instead of "Ref. [10]" in the legend
6) The paper "Topological Elasticity of Nonorientable Ribbons" https://journals.aps.org/prx/abstract/10.1103/PhysRevX.9.041058 by Bartolo and Carpentier might interest the authors and perhaps be a useful reference.

---

## Editorial Decision

unknown